# Performance Comparison of CdTe:Na, CdTe:As, and CdTe:P Single Crystals for Solar Cell Applications

**DOI:** 10.3390/ma15041408

**Published:** 2022-02-14

**Authors:** Sangsu Kim, Deok Kim, Jinki Hong, Abdallah Elmughrabi, Alima Melis, Jung-Yeol Yeom, Chansun Park, Shinhaeng Cho

**Affiliations:** 1Department of Display and Semiconductor Physics, Sejong Campus, Korea University, 2511 Sejong-ro, Sejong City 30019, Korea; kss2193@korea.ac.kr (S.K.); jkhongjkhong@korea.ac.kr (J.H.); 2Department of Radiation Oncology, Chonnam National University Medical School, 160 Baekseo-ro, Gwangju 61469, Korea; kupid05@naver.com; 3Department of Bio-Microsystem Technology, Korea University, 145 Anam-ro, Seoul 02841, Korea; iglobalthoughts@gmail.com; 4School of Biomedical Engineering, Korea University, 145 Anam-ro, Seoul 02841, Korea; alima28@korea.ac.kr (A.M.); jungyeol@korea.ac.kr (J.-Y.Y.); 5BK21 Four R&E Center for Precision Public Health, Korea University, 145 Anam-ro, Seoul 0284, Korea

**Keywords:** CdS/CdTe, solar cell, single crystal, crystal growth, thermal stability, vertical Bridgman technique

## Abstract

We compared thermal stability, open-circuit voltage, short-circuit current, and fill factor values of single-crystal Cadmium telluride (CdTe) grown using the vertical Bridgman (VB) technique and doped with group V elements (phosphorus and arsenic), and group Ⅰ element (sodium), followed by an annealing process. The sodium-doped CdTe maintained a hole density of 10^16^ cm^−3^ or higher; after annealing for a long time, this decreased to 10^15^ cm^−3^ or less. The arsenic-doped CdTe maintained a hole density of approximately 10^16^ cm^−3^ even after the annealing process; however its bulk minority carrier lifetime decreased by approximately 10%. The phosphorus-doped CdTe maintained its properties after the annealing process, ultimately achieving a hole density of ~10^16^ cm^−3^ and a minority carrier lifetime of ~40 ns. The characteristics of a single-crystal solar cell were evaluated using a solar cell device that contained single-crystal CdTe with various dopants. The sodium-doped sample exhibited poor interfacial properties, and its performance decreased rapidly during annealing. The samples doped with group V elements exhibited stable characteristics even during long-term annealing. We concluded, therefore, that group V elements dopants are more suitable for CdTe single-crystal-based solar cell applications involving thermal stress conditions, such as space missions or extreme fabrication temperature environments.

## 1. Introduction

CdTe-based photovoltaic (PV) cells, one of the most commercially successful solar energy harnessing technologies, are used for producing low-cost and high-efficiency solar panels. Consequently, they have been studied extensively for applications related to solar-cell-based devices, X-ray detectors, and other similar devices. Conventional polycrystalline CdTe solar cells are approximately 22% efficient as a result of optimizations such as increasing the grain size, open-circuit voltage (V_oc_), short-circuit current (J_sc_), and fill factor (FF) values [1]. However, this efficiency is still below the Shockley–Queisser limit [2,3], and thus, various methods have been proposed [4,5] to further improve the performance of CdTe solar cells. The properties of CdTe materials and their effects on CdTe PV cells were evaluated in a recent study [6,7]. For CdS- and CdTe-based devices, J_sc_ has already been optimized; for an optimal J_sc_, the total carrier lifetime should be a few nanoseconds with a hole density in the range 10^13^ to 10^15^ cm^−3^ [8,9]. Duenow et al. [10] fabricated CdTe cells doped with Na (a group I dopant), exhibiting hole densities exceeding 5 × 10^15^ cm^−3^ and V_oc_ exceeding 900 mV. Metzger et al. [11] fabricated a single-crystal CdTe phosphorus (CdTe:P) cell by doping the anionic site with P (a group V element) to overcome the limitations of CdTe materials. They maintained the V_oc_ between 840 and 880 mV and obtained a hole density exceeding 10^16^ cm^−3^. Reportedly, a hole density of 10^16^ cm^−3^ and V_oc_ of more than 1000 mV can be achieved [12]. Furthermore, considering the increasing applications of CdTe-based cells, apart from efficiency, their long-term stability and applications in high-radiation environments, such as space missions, need to be assessed as well. Cho et al. [13] fabricated a CdTe-based cell that can be utilized for space missions or beam-induced treatments and investigated the degradation process of the cell in a proton beam environment. Burst et al. [14] showed that the hole density and carrier lifetime of CdTe can change when the sample is annealed in an environment containing Cd or Te vapor.

However, to fabricate an optimal cell that can sustain its performance under various stress conditions would require several materials to be experimentally reviewed in order to identify the most suitable dopant. In addition, a material suitable for long-term usage would need to be stable even when exposed to an excessively stressful environment. In this study, CdTe single crystals doped with group I (Na) and group V dopants (P and As) were grown in the form of ingots using the vertical Bridgman (VB) technique. By using single-crystal CdTe, the effect due to grain boundaries of polycrystalline CdTe could be eliminated and the characteristics according to the type of dopant could be observed more clearly. Their properties were evaluated under thermal stress with a focus on their carrier lifetimes and hole densities, as these are important parameters for improving the V_oc_. The initial carrier lifetime and hole density may vary owing to various factors such as the growth method, type of dopant, doping level, stoichiometry, impurities, secondary phase inclusions, and other process variations [15,16,17,18]. Therefore, the experimental objective was achieved using the VB technique and by varying the dopant. The annealing test data of a single-crystal sample included the changes resulting from exposure to high temperatures for a short period of approximately 30 min and long-term thermal stress for up to 10 days. Thereafter, we fabricated individual solar cells with single-crystal CdTe and added the dopants to each of them. This single-crystal CdTe PV device contained a CdS/ZnO window layer and Cu/Au bilayer back contacts. Utilizing Cu in CdTe PV devices is a conventional technology that has played an important role in the formation of efficient back junctions, despite the ability of Cu to easily change energy states and diffuse spatially. Since we want the device to operate in an environment such as in space or near a nuclear reactor, the fabricated device was stress tested at a temperature of around 100 °C. Korevaar et al. [19,20] and other researchers reported that utilizing Cu could cause cell instability during long-term usage. In this study, the final device was fabricated after annealing the single crystals in a long-term thermal stress environment to prevent the Cu from being incorporated into the cell junction. In addition, identical annealing was performed on the final cell manufactured in the initial state for comparison; the comparison of the efficiencies of the two devices indirectly included the effects of the presence and absence of Cu.

## 2. Materials and Methods

### 2.1. Single-Crystal CdTe Growth Using the VB Technique

The VB technique was used to grow a single crystal of CdTe by using Cd and Te of 7N purity purchased from Nippon Mining & Metals Co. Ltd. (Tokyo, Japan). As dopants, we used Cd_3_As_2_ (CAS No. 12006-15-4), Cd_3_P_2_ (CAS No. 12014-28-7), and Na_2_Te (CAS No. 12034-41-2). The quantities of Cd and Te were fine-tuned based on the quantity of doping material. An ampoule manufactured from GE quartz was used for the crystal growth. The shape of the ampoule was unified as a cone, and its inner surface was coated with graphite to prevent any potential reactions during the growth process. Furthermore, to prevent residue mixing, the ampoule was cleaned with aqua regia, hydrofluoric acid, and deionized water. Thereafter, it was washed with a solution containing MeOH, acetone, and trichloroethylene (TCH) (MeOH–acetone–trichloroethylene–acetone–MeOH) and dried. Gases were removed from the ampoule by placing it under a vacuum of ~10^−7^ Torr, after which it was sealed with a 2” quartz plug. Crystal growth proceeded at a rate of approximately 1.5 mm/h. To prevent distortions during the growth of CdTe a convective melt mixing (soaking) process was used at the melting-point temperature of Cd and Te, and the CdTe was formed at a temperature of approximately 1150 °C. After crystal growth, the ingots were cooled at a cooling rate of 5–10 °C/h, and the resulting single-crystal CdTe ingots were cut using a diamond saw. Lastly, the cut CdTe wafer was mechanically polished and chemically etched in a solution of 1% BrMe. Images of the grown ingots and polished samples are presented in Figure 1.

### 2.2. Solar-Cell Device Fabrication

The final wafer size and thickness of the fabricated solar-cell devices were 8 mm × 8 mm and approximately 800 μm, respectively. The thickness of the sample was fine-tuned by adjusting the etching time using the BrMe solution and measured using a micrometer. Considering the yield when manufacturing solar cell devices, we selected the sample with the thinnest thickness that could be controlled. For electrode formation, the edges were masked with Kapton tape, and the unmasked area was defined as the active cell area. To form a solar cell structure (Figure 1b), a 100–150 nm CdS layer and a 150 nm ZnO layer were sequentially deposited on the sample using radiofrequency (RF) magnetron sputtering, with a base pressure of 10^−7^ Torr inside the sputtering chamber. To form the back-contact electrode, Cu/Au was deposited on the reverse side of the sample through direct current (DC) magnetron sputtering. The thicknesses of the Cu and Au layers were ~4 and ~150 nm, respectively. The contact between the front and back electrodes was formed by a contact method using an In press and a 40-μm-thick Au wire.

### 2.3. Measurement Details

The hole density was obtained from the Hall-effect measurements performed using the Van der Pauw technique. The carrier lifetime was determined by the two-photon excitation time-resolved photoluminescence (2PE-TRPL) technique. The samples were excited at 1100 nm using a laser/optical parametric amplifier with a repetition rate, pulse length, average power, and spot size of 1 × 10^5^ pulses/s, 0.35 ps, 7–20 mW, and ~20 µm, respectively. The PL emission was detected at 840 nm using a 10 nm bandpass filter. We ensured reliability of the data by taking measurements at several points on the sample. Compared with the PL measurements in the single-photon mode, the obtained two-photon excitation lifetimes were relatively insensitive to the surface recombination effects; therefore, we could effectively measure the bulk recombination effect. The current density–voltage (J–V) characteristics were evaluated at an illumination of 100 mW cm^−2^ under AM 1.5 conditions. A glow discharge mass spectroscopy (GDMS) measurement was performed to determine the impurity concentration level of the doped CdTe ingot. In order to observe the distribution of impurities according to the location of the ingot, measurements were carried out using samples near the top and bottom of the ingot. Table 1 lists the results.

## 3. Results and Discussion

### 3.1. Single-Crystal CdTe

We investigated the carrier lifetime of single-crystal CdTe containing a group I cationic dopant (Na) and group V anionic dopants (P and As). Figure 2 shows the carrier lifetime of the material as determined by the 2PE-TRPL method. The single crystals containing each type of dopant were measured as grown samples and annealed at 200 °C and 400 °C. All annealing processes were performed in a nitrogen environment of 1 atm. Initially, in the as-grown state, the CdTe:P crystal exhibited a carrier lifetime of 40–47 ns. After the crystal was annealed at 200 °C for 30 min, the carrier lifetime decreased to 39–44 ns, whereas after annealing at 400 °C, the measured carrier lifetime was 38–44 ns. The CdTe:As crystal initially exhibited a carrier lifetime of 40–45 ns in the as-grown state. After annealing the crystal at 200 and 400 °C, the respective carrier lifetimes were shortened to 36–40 ns and 35–40 ns, respectively. These results indicated that the carrier lifetimes of the CdTe samples doped with P and As were shortened by approximately 5–10% after the annealing process. However, no substantial change was observed between the samples annealed at 200 and 400 °C. The initial carrier lifetimes of P and As were both approximately 42 ns, which was nearly identical to those of the CdTe single crystals doped with group V elements reported elsewhere [21]. In the as-grown state, the CdTe:Na crystal exhibited an initial carrier lifetime of 8–10 ns, which after annealing at 200 °C, increased slightly to 8.6–11 ns. However, after annealing at 400 °C, the lifetime decreased to 4.8–8.6 ns. The carrier lifetime of the CdTe single crystals ranged from several nanoseconds to several tens of nanoseconds, exhibiting a fairly large gap between those shown by the single crystals doped with group I (Na) and with group V (P and As) elements. However, compared with the reported lifetime of 1–2 ns for the polycrystalline CdTe film devices, the overall values obtained in this study were very large [22].

Table 2 lists the measured carrier lifetimes and hole densities for each sample in the as-grown and post-annealing (annealed at 400 °C) states. These properties represent the main parameters that determine the V_oc_ of the fabricated solar-cell devices [23]. These values were obtained by averaging those obtained from three samples of each doped single crystal. The hole density of the CdTe single crystals in the as-grown state was 10^15^–10^16^ cm^−3^, which is approximately ten times higher than those obtained in a typical polycrystalline CdTe/CdS device (approximately 10^14^–10^15^ cm^−3^) [24]. According to the results in Table 2, annealing the CdTe single crystals doped with group V dopants (P and As) at 400 °C for 30 min increased the hole density by more than two times. In contrast, for the CdTe:Na crystal, the hole density decreased by approximately 10 times after the same annealing treatment.

To evaluate variations in the hole density, we annealed all the samples at each temperature. Figure 3a displays the change in the hole density of the annealed crystal at each temperature. Evidently, the P- and As-doped crystals exhibited stable characteristics, that is, their respective hole densities did not change significantly at temperatures in the vicinity of 250 °C. As the temperature increased, the hole density of CdTe:As began to increase to approximately 300 °C and gradually increased up to 400 °C. After annealing at 400 °C for 30 min, the hole density of CdTe:P was almost unchanged and the hole density of CdTe:As increased by approximately 1.2 times. As the growth conditions for each sample were the same, the changes in the hole density could only be due to the variations in the annealing temperature. These results revealed that a minimum annealing temperature of 300 °C was required to move the dopant atoms from the interstitial sites to the substitution sites in the lattice to enable p-type doping in the CdTe:P and CdTe:As crystals. Therefore, assuming that the annealing temperature for manufacturing a typical solar cell device is between 200 and 400 °C, the use of an annealing temperature of 300 °C or higher before the cell construction is likely to be advantageous for CdTe-based solar-cell devices (doped with group V dopants: P and As). Moreover, the results for CdTe:Na were in contrast with those for the samples doped with group V dopants. In the as-grown state, the CdTe:Na single crystal could achieve a hole density of approximately 10^16^ cm^−3^, which was unique to this crystal. However, after undergoing annealing at 150 °C, its hole density decreased slightly, and from 200 °C onward, it decreased significantly. In the vicinity of 250 °C, the hole density decreased to 10 times lower than its initial value. These results indicated that the Na-doped single crystal could exhibit stability problems during the solar-cell device manufacturing process. This was consistent with the results of a previous study, which showed that group I elements, Na and Cu, could cause problems in the general solar-cell manufacturing process [25]. In addition, group I dopants could cause stability problems owing to bulk-defect chemistry instead of the grain boundary effects [26].

The results of the long-term thermal stability test of the single crystals are presented in Figure 3b. In this case, the temperature was fixed at 100 °C, and the samples were annealed for up to 10 days. The results showed a substantial change in the single-crystal sample doped with Na. The initial hole density of the sample (10^16^ cm^−3^) continued to decrease to eventually reach a value of 5 × 10^14^ cm^−3^. This value was similar to or lower than those reported for typical polycrystalline devices. The aforementioned results on hole density and carrier lifetime revealed that manufacturing solar-cell devices with group I dopants, including Na, might present substantial stability issues.

### 3.2. Solar-Cell Devices

To evaluate the compatibility of the doped CdTe samples, a solar-cell device was fabricated from the CdTe single-crystal cell prepared using the VB technique. The face bond was fabricated on the mechanically polished crystals using a conventional method. Next, a CdS/ZnO window layer was sequentially deposited on the grown CdTe single crystals, and Cu/Au contacts were formed as the back-contact electrode. In a previous experiment, we demonstrated the negative effects of Na, which is a group I dopant; a similar effect was expected from the use of Cu as the back-contact electrode. As careful attention was required for fabricating the single-crystal cells using Na, the annealing treatment for sample activation was divided into two steps. After the front bonding, we annealed the sample at 400 °C to activate it. Next, the sample was turned over and etched in an NP (i.e., a mixture of phosphoric acid and nitric acid) solution for 1 min to lower the back-contact resistance. The back-contact electrode was deposited using the DC magnetron sputtering process and activated using short-cycle annealing at 250 °C for 2 min to minimize the influence of Cu. All the deposition processes were performed at room temperature. The performance of the finished PV device was evaluated at room temperature under AM 1.5 conditions (100 mW·cm^−2^ intensity). To test the long-term thermal stability of the sample, the measurements were conducted at room temperature after annealing the sample at 100 °C for 10 days.

The J–V curve of the sample is shown in Figure 4, and the performance factors of the fabricated solar cells are listed in Table 3. Evidently, all the devices prepared from the single crystals, doped with each type of dopant, were able to perform efficiently at room temperature, exhibiting stable characteristics. In addition, at room temperature, the V_oc_ of the solar-cell device was between 800 and 860 mV, and the J_sc_ was between 23 (for P and As dopants) and 20 mA (for Na). Next, each sample was annealed at 100 °C for 10 days, and the measurements were repeated. After annealing, the overall performance of each device deteriorated. In the case of P- and As-doped samples, the performances of the two devices were nearly identical; their V_oc_ and J_sc_ decreased to approximately 780 mV and 18 mA, respectively, which could be linked to the use of Cu for the back contact as Cu can diffuse from the back contact surface and affect the performance of the device. This observation was consistent with those reported for polycrystalline CdTe solar cells [19,20]. In addition, it has been reported that Cu can cause problems in the long-term stability of CdTe [25,27,28]. In contrast, although the overall performance of the solar-cell device containing P- and As-doped single-crystal CdTe deteriorated, the FF of the individual devices remained approximately constant. As the sample thickness was approximately 800 μm, it could be assumed that, during the annealing process, Cu did not diffuse to the front junction region. For the Na-doped device, the cell performance deteriorated significantly, which was consistent with the characteristics of group I dopants Cu and Na. This degradation in performance was possibly due to the penetration of the p–n junction by elements with large diffusion coefficients. As a result, the PV devices with Na and back Cu junctions exhibited a significant decrease in V_oc_, J_sc_, and FF. Between group I and group V dopants, the latter exhibited desirable thermal stabilities. However, Cu, which is a conventionally used group I element in the CdTe device manufacturing process, was still considered to negatively affect the device’s stability. Therefore, further research is necessary to overcome the shortcomings of Cu in the manufacturing of single-crystal cells and to develop a process that does not require Cu.

## 4. Conclusions

In this study, individual CdTe single crystals doped with Na (group I dopant) as well as P and As (group V dopants) were grown using the VB technique. To optimize the fabrication of PV devices, we investigated the change in hole density, V_oc_, J_sc_, and FF for each dopant during annealing. The results indicated that the samples doped with P and As exhibited a hole density of approximately 10^16^ cm^−3^ after annealing. This value was maintained on the device with a surface area of 9 × 10^15^ cm^−2^ during long-term annealing for 10 days. For the Na-doped sample, the hole density decreased by more than 10 times its initial value because of the diffusion effect; additionally, the stability of this sample was unsatisfactory.

To evaluate the performance of the solar-cell devices prepared using the single-crystal samples, the grown single crystals were fabricated with CdTe/CdS structures. According to the report of J.M. Burst et al. [12], it is possible to fabricate a device with an open-circuit voltage of 1 V and a short-circuit current density of 25 mA/cm^2^ using single-crystal P-doped CdTe. Our device had an open-circuit voltage of about 0.87 V and a short-circuit current density of 24 mA/cm^2^. This suggested that a more delicate process was needed in terms of performance optimization. The front window layer and the Cu-containing back electrode were deposited using DC and RF magnetron sputtering, respectively, and their cell performance and stability were compared. The stability was evaluated after annealing the samples for 10 days. For the samples containing group V dopants, an overall decrease in the cell efficiency was observed after annealing because of the decrease in V_oc_ and J_sc_. This performance degradation was believed to be related to the diffusion of Cu from the back contact into the cell layers and junctions. The Na-doped sample underwent more extensive degradation compared to the samples containing group V dopants, possibly because of the increase in surface recombination due to the diffusion of Na and Cu into the junction region. Based on these results, we can conclude that group V dopants are more suitable for CdTe single-crystal-based solar cells that are used in applications subjected to thermal stress, such as space missions, extreme fabrication temperature environments, etc. In the future, we plan to study ways to resolve the shortcomings of Cu in the manufacturing of single-crystal cells or developing a process that does not utilize Cu.

## Figures and Tables

**Figure 1 materials-15-01408-f001:**
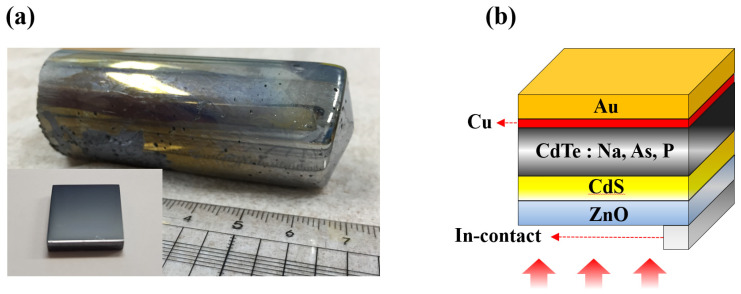
(**a**) Grown ingot and polished sample (inset); (**b**) schematic diagram of the device structure.

**Figure 2 materials-15-01408-f002:**
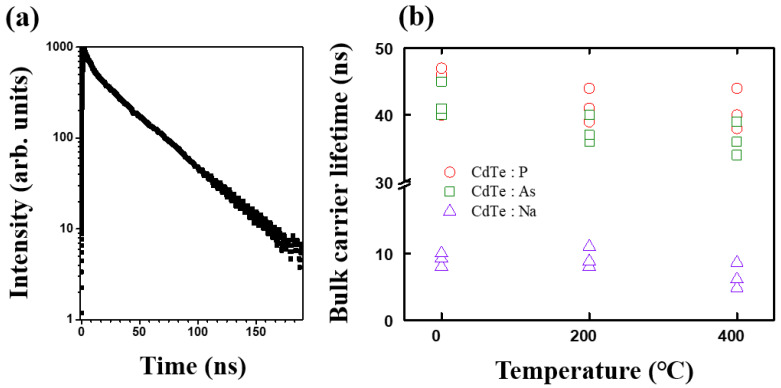
(**a**) 2PE-TRPL decay curve for CdTe:P; (**b**) bulk carrier lifetimes for the CdTe single crystals with each of the dopants annealed at each temperature. The lifetimes were calculated from the 2PE-TRPL measurements.

**Figure 3 materials-15-01408-f003:**
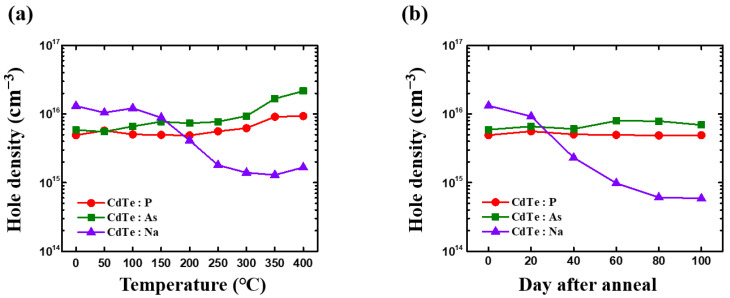
(**a**) Changes in the hole density of CdTe single-crystal samples annealed for 30 min at each temperature; (**b**) changes in the hole density of CdTe single-crystal samples annealed at a temperature of 100 °C for ~100 days.

**Figure 4 materials-15-01408-f004:**
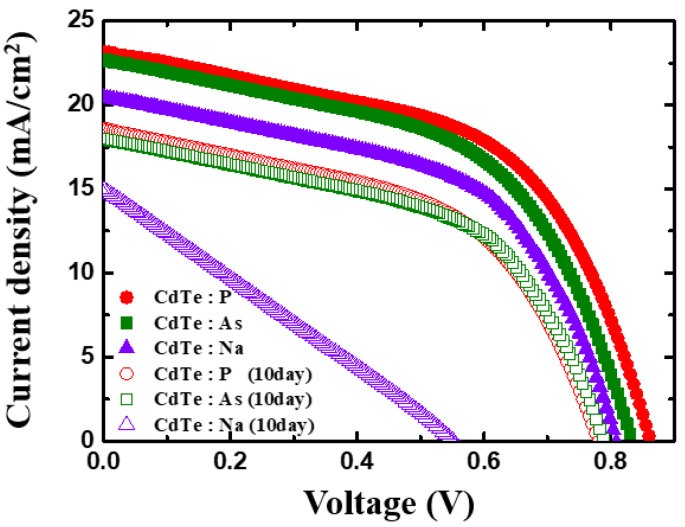
J–V curve of CdTe:P, CdTe:As, and CdTe:Na solar-cell devices. The solid symbols represent the as-deposited samples, and the hollow symbols represent the samples annealed at 100 °C for 10 days.

**Table 1 materials-15-01408-t001:** Impurity levels in the doped ingots determined by GDMS analysis. Data for CdTe:P, CdTe:As, and CdTe:Na are shown in order from left. The units are PPB.

Elements	Top	Bottom	Elements	Top	Bottom	Elements	Top	Bottom
C	400	310	C	410	310	C	390	300
N	20	20	N	20	20	N	20	20
O	700	810	O	720	790	O	700	710
Na	20	140	Na	30	130	Na	7100	8000
Mg	40	30	Mg	40	30	Mg	40	30
Al	10	10	Al	10	10	Al	10	10
Si	10	20	Si	10	20	Si	10	20
P	17,000	18,000	P	80	60	P	80	60
S	30	30	S	30	30	S	30	30
Cu	40	80	Cu	30	70	Cu	30	70
As	40	50	As	22,000	23,000	As	40	50

**Table 2 materials-15-01408-t002:** Parameters of the CdTe:P, CdTe:As, and CdTe:Na materials in the as-grown and annealed (400 °C, 30 min) conditions.

Samples	Hole Density (cm^−3^)	Carrier Lifetime (ns)
CdTe:P	4.92 × 10^15^	40–47
CdTe:P (after annealing)	9.34 × 10^15^	39–42
CdTe:As	5.84 × 10^15^	40–45
CdTe:As (after annealing)	2.16 × 10^16^	34–39
CdTe:Na	1.01 × 10^16^	9–13
CdTe:Na (after annealing)	1.68 × 10^15^	4.8–9.6

**Table 3 materials-15-01408-t003:** Performance parameters of CdTe:P, CdTe:As, and CdTe:Na solar-cell devices.

Samples	V_oc_ (V)	J_sc_ (mA/cm^2^)	FF (%)	Eff. (%)
CdTe:P	0.86	23.3	53.6	10.7
CdTe:As	0.83	22.6	53.2	10.0
CdTe:Na	0.81	20.6	53.1	8.8
CdTe:P (10 day)	0.78	18.8	50.9	7.5
CdTe:As (10 days)	0.79	17.9	52.1	7.4
CdTe:Na (10 days)	0.54	15.1	25.9	2.1

## Data Availability

Data underlying the results presented in this paper are not publicly available at this time but may be obtained from the authors upon reasonable request.

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
