# Peer review of "Performance Comparison of CdTe:Na, CdTe:As, and CdTe:P Single Crystals for Solar Cell Applications"

_materials, 2022, doi:10.3390/ma15041408_

Round 1
Reviewer 1 Report
The article presents materials on crystal growth of CdTe:Na, CdTe:As, CdTeLP compounds and the study of its parameters which useful for solar cells. Special attention is paid to the temperature treatment of crystals and solar cells In principle, useful materials are presented, but they need to be improved
- the purpose of the work should be clarified.What did the authors want to improve in comparison with the known methods of creating solar cells
- it is necessary to characterize the defective structure of crystals in some way - first of all, the distribution of dopants
- Correct table 1 of the content that does not answer the title
- In the conclusions, it is necessary to compare the parameters of the developed solar cells with the known ones
Reviewer 2 Report
The research manuscript ID: materials-1570112 entitled "Performance comparison of CdTe:Na, CdTe:As, and CdTe:P single crystals for solar cell applications" submitted to Journal of Materials describes synthesis, optimization and the individual CdTe single crystals doped with Na (Group I dopant) as well as P and As (Group V dopants) grown using the VB technique.
The authors just performed good analysis for this study, and the results seem significant. Though, it needs minor revision before it can be accepted. Some of the corrections and suggestions are as follows:
- In the abstract the authors must check the exponent of (1015, 1016).
- In figure 1 the authors must specify the Cu electrode in Schematic diagram of the device structure.
- In section 2.2 the authors should indicate how they measured the thickness of the final wafer and specify the choice of this thickness.
Reviewer 3 Report
The authors present a performance comparison of CdTe single crystals doped with Na, As, and P to judge their possible applicability for solar cell devices. The results are presented well, however, there are some smaller issues that have to be addressed before a possible publication in the journal Materials. Please find my detailed comments below.
Abstract.
Please correct the hole densities, i.e. instead of “1016 cm-3” (line 20) write “10^{16} cm^{-3}”. There are more occurrences (line 21, 22, and 25).
Keywords.
Maybe it’s worth adding “vertical Bridgman technique” to the list of keywords.
1. Introduction.
When you state the efficiency of CdTe solar cell devices of about 22 %, please add a reference. If that’s not possible, cite the NIST best research-cell efficiency chart (https://www.nrel.gov/pv/cell-efficiency.html).
3. Results and discussion.
On page 5, line 182, you state that hole densities are approximately 100 % and 120 %. This sentence is misleading (100 % compared to what), please find a better description.
I’d suggest a line break on page 5, line 202, just before the start of the long-term thermal stability test. Is there a specific reason why you fixed the annealing temperature to 100 °C for the long-term thermal stability test? I’m asking, since previously you mentioned that annealing temperatures of 300 °C would be advantageous. Please add a comment.
Table 2 on page 7 is exactly the same as Table 1 on page 4. Please correct this.
Author contributions.
Please remove the sentences from the template and leave only the real author contributions. Here, it might also be worth to collect the abbreviations used in the manuscript.
References.
The references need to be checked and corrected, i.e. for some of them their number is given twice.
Reviewer 4 Report
Within the paper Performance comparison of CdTe:Na, CdTe:As, and CdTe:P single crystals for solar cell applications authors compare three CdTe substrates doped with different dopants: sodium Na, phosphorus P and arsenic As. The idea of paper is very interesting especially for researchers related with crystal growth and photovoltaics. However, the present version of the paper cannot be published in Materials. It has to be carefully revised. Bellow I listed my detailed remarks:
- abstract - please use commonly used notation for free carrier concentration i.e. 1015 cm-3 instead of 1015 cm-3 - see line 47 where this is correct
- abstract - where authors describe carrier lifetime please specify they mean majority or minority carriers
- Annealing of fabricated CdTe samples - there is no information about important annealing parameter i.e. atmosphere. Was it air, nitrogen, hydrogen or Te- or Cd-rich vapour?
- Materials and methods - please add information about dopant sources producer and if possible CAS numbers of used materials
- Figure 1b - there is no description of thin red layer in the figure, is it Cu?
- Results and discussion - authors claims that carrier lifetime was determined on the basis of 2PE-TRPL measurements, but in the main text the original results of PL decay are not shown - please add some results
- lines 138-139: The single crystals containing each type of dopant were measured at three temperatures: 0, 200, and 400 °C. I would recommend to change the sentence to The single crystals containing each type of dopant were measured for as grown samples and annealed at 200 and 400C.
- Table 1 is the same as Table 2 so the description in lines 165-166 is not true
- Generally, when some data concerning monocrystalline wafers doped with different dopant are published authors investigate the electrical properties by ECV methods. This approach yield in carrier concentration vs the depth of the wafer/sample. In my opinion ECV results will significantly enhance the quality of the paper where 3 different dopants are compared
- lines 244-245 and 249-251. To be sure about Cu interdiffusion process induced by device processing and 10-days annealing it would be interesting to investigate both samples (as fabricated and after 10-days annealing) by SIMS. This measurement usually dispel doubts about any diffusion across interfaces in the investigated structure
Round 2
Reviewer 1 Report
The reviewer"s comments are taken into account and paper could be published
Reviewer 4 Report
In my opinion the quality of revised paper is sufficient for publication in Materials.